# Synthesis and Antimicrobial Activity Screening of Piperazines Bearing *N*,*N′*-Bis(1,3,4-thiadiazole) Moiety as Probable Enoyl-ACP Reductase Inhibitors

**DOI:** 10.3390/molecules27123698

**Published:** 2022-06-09

**Authors:** Alaa Z. Omar, Najla A. Alshaye, Tawfik M. Mosa, Samir K. El-Sadany, Ezzat A. Hamed, Mohamed A. El-Atawy

**Affiliations:** 1Chemistry Department, Faculty of Science, Alexandria University, P.O. 426 Ibrahemia, Alexandria 21321, Egypt; mhmwdt18@gmail.com (T.M.M.); samir.elsadany@alexu.edu.eg (S.K.E.-S.); ezzat.awad@alexu.edu.eg (E.A.H.); 2Department of Chemistry, College of Science, Princess Nourah Bint Abdulrahman University, P.O. Box 84428, Riyadh 11671, Saudi Arabia; naalshaye@pnu.edu.sa; 3Chemistry Department, Faculty of Science, Taibah University, Yanbu 46423, Saudi Arabia

**Keywords:** enoyl reductase, MOE, 1,3,4-thiadiazole, piperazine, 1,2,4-triazole

## Abstract

A new *N*,*N′*-disubstituted piperazine conjugated with 1,3,4-thiadiazole and 1,2,4-triazole was prepared and the chemical structures were identified by IR, NMR and elemental analysis. All the prepared compounds were tested for their antimicrobial activity. The antimicrobial results indicated that the tested compounds showed significant antibacterial activity against gram-negative strains, especially *E. coli*, relative to gram-positive bacteria. Docking analysis was performed to support the biological results; binding modes with the active site of enoyl reductase amino acids from *E. coli* showed very good scores, ranging from −6.1090 to −9.6184 kcal/mol. Correlation analysis was performed for the inhibition zone (nm) and the docking score.

## 1. Introduction

There is an urgent need to develop new generations of antimicrobial drugs with distinct modes of actions [1,2]. The discovery of new antimicrobial drugs is now focused on drug targets like enzymes or receptors. The bacterial enoyl acyl carrier protein reductase (ENR) is one of the attractive targets for the discovery of antibacterial drugs [3] due to its vital role in fatty acid biosynthesis and its sequence conservation across many bacterial species. Moreover, the bacterial ENR structural organization and sequence are markedly different from those of human or mammalian fatty acid biosynthesis [4]. ENR is utilized in the last step of the elongation cycle in the biosynthesis of fatty acids, which comprise the reduction of enoyl-acyl carrier proteins to the fully saturated acyl-acyl carrier protein species. 

The reduction of the double bond occurs through conjugate addition of a hydride ion to beta carbon with respect to the acyl thioester moiety. The hydride is derived from the coenzyme nicotinamide adenine dinucleotide (NADH or NADPH). The hydride transfer process leads to the formation of an enzyme-stabilized enolate anion intermediate. Subsequent protonation of the enolate by a proton from the hydroxyl group of tyrosine side chain (Tyr 158) produces fatty acids. The Tyr 158 proton is regenerated from the solvent (water) through a protonation–deprotonation system between the amino group of Lys 163 and the hydroxyl groups of the coenzyme ribose moiety in addition to water molecules. Thus, the reduction process is ruled by the active sites of the enzyme (two key residues Tyr 158 and Lys 163) in addition to the NAD(H) coenzyme present in the enzyme active site [5]. Other moieties, such as triazoles, thiadiazoles, and piperazines are also reported to possess inhibition towards the previous enzyme target [6,7,8]. 

Piperazine derivatives have a wide range of biological activities [9,10,11] owing to the basicity of their nitrogen atoms and the electrostatic interaction between the protonated piperazine and the anionic region of the receptor. The varying of *N*-substituents on piperazine have an important role in the potency towards the biological targets [12] as well as improved selectivity of the molecule [13].

1,3,4-Thiadiazole derivatives have been known to possess antimicrobial [14,15,16,17,18,19,20,21,22,23,24], antitubercular [25], anti-helicobacter pylori [26], antituberculosis [27,28], antioxidant [29,30], anti-inflammatory [31,32,33], anticonvulsants [34], antidepressant and anxiolytic [35], antihypertensive [36], anticancer [37,38,39,40,41,42,43], and antifungal activity [44,45,46,47]. 1,3,4-Thiadiazoles also exhibit high potential in agriculture, such as in pesticides, herbicides, fungicides, insecticides, bactericides, and even plant-growth regulators. It was reported that the impact on antibacterial activity of *N*-substituted piperazinyl carrying benzylthio-1,3,4-thiadiazole moiety inhibited the formation of the 70S initiation complex, which is responsible for protein biosynthesis. This compound was found to be more potent than ciprofloxacin against a panel of gram-positive and gram-negative bacteria [48,49]; see Figure 1. 

Owing to the above facts and in continuation of the research on enoyl ACP reductase inhibitors, an attempt was made in the present research to design and develop new anti-*E. coli* agents possessing enoyl ACP reductase inhibition. The extensive biological activity of *N*,*N*′-bis substituted piperazine molecules prompted us to functionalize and design novel similar compounds containing *N*,*N*′-bis(1,3,4-thiadiazole) and 1,2,4-triazole moieties [50,51,52] to give a confined structure for evaluating the antimicrobial activities [53]. The design strategy is presented in Figure 1. The docking analysis for the active sites of enoyl reductase from *E*. *coli* are compared with the experimental biological results.

## 2. Results and Discussion

### 2.1. Chemistry

The synthesized 5,5′-(piperazin-1,4-diyl)bis(1,3,4-thiadiazole-2-thiol) **1** [54] was selected as a precursor to give 1,4-bis(5-hydrazinyl-1,3,4-thiadiazol-2-yl)piperazine **2** and 1,4-bis(5-chloro-1,3,4-thiadiazol-2-yl)piperazine **3** by its reaction with 98% hydrazine hydrate [55] and thionyl chloride in benzene [56], respectively; see Figure 1.

Obviously, the synthesis of new scaffolds of 1,2,4-triazoles fused with 1,3,4-thiadiazole are relevant and appropriate, owing to their significant biological activity [57]. Refluxing of 1,4-bis(hydrazinyl)piperazine derivative **2** in acetic anhydride led to the formation of 1,4-bis(3-methyl-1,2,4-triazolo[3,4-*b*]-1,3,4-thiadiazol-6-yl) piperazine **4** in excellent yield (87%); see Figure 2. Whereas, heating of compound **2** with carbon disulfide in pyridine furnished 6,6′-(piperazin-1,4-diyl)bis(1,2,4-triazolo[3,4-*b*]-1,3,4-thiadiazole-3-thiol) **5** in a yield of (67%); see Figure 2.

The ease of replacement of the chloro atom in compound **3** enabled us to make further functionalization of piperazine containing the 1,3,4-thiadiazole system. The replacement of the chloro atom in compound **3** by substituting thiophenoxides (C_6_H_5_, 4-NO_2_C_6_H_4_, 4-BrC_6_H_4_, 4-CH_3_C_6_H_4_) gave compounds **6a–d** in moderate to good yield, depending on the nature of the substituent; see Figure 3. Moreover, refluxing the methanolic solution of piperazine derivative **3** with 2-mercaptobenzoic acid or 2-mercaptonicotinic acid in the presence of a few drops of triethyl amine yielded piperazine bisbenzoic **7a** and piperazine bisnicotinic acids **7b,** respectively. Furthermore, the reaction of compound **3** with two folds of thioglycollic acid yielded compound **8;** see Figure 3. Furthermore, aniline and morpholine react with compound **3** in the presence of triethyl amine, giving **9** and **10**, respectively; see Figure 3.

All of the compounds that were synthesized as a result of this study were able to withstand exposure to air for a long period of time and could be dissolved in typical solvents such as DMF and DMSO. The infrared (IR) spectra of the piperazine ring of compounds **2**–**10** showed two main vibration peaks. The first peak in the range υ 2983–2898 cm^−1^ was attributed to the stretching vibration of the aliphatic C-H group, whereas the second vibration band appeared in range of υ 1190–1093 cm^−1^, assigned to the vibration of the C-N bond. Moreover, the thiadiazole ring was characterized by two main absorption peaks: the imino group (C=N), observed as medium peaks at υ 1637–1554 cm^−1^, and the thioether (C-S) bond, which appeared as a weak signal in the range of υ 717–744 cm^−1^ [58] Additionally, compound **2** had two weak peaks at υ 3296 and 3146 cm^−1^, attributed to the N-H stretching vibration. Compounds **6a–d** exhibited a weak vibration band in the range of ν 3025–3094 cm^−1^, attributed to the stretching vibration of the aromatic C-H group. Furthermore, **6b** possessed two medium peaks at ν 1510 and 1339 cm^−1^, corresponding to the asymmetric and symmetric stretching vibration of the nitro substituent. The O-H stretching vibration of the carboxylic group of compounds **7a,b** and **8** appeared as broad peaks in range of ν 3436, 3437, and 3441, respectively. Their carbonyl group appeared as a sharp strong peak at ν 1664, 1681, and 1630 cm^−1^, respectively. Moreover, anilino derivative **9** possessed an additional vibrational peak at ν 3443 and 3074 cm^−1^, corresponding to the stretching of NH and aromatic C-H bonds, respectively.

The proton Nuclear Magnetic Resonance (NMR) spectra of the symmetrical structure of compounds **2**–**10** exhibited only one singlet signal in the range of δ 3.77–3.15 ppm for the piperazine protons. This also was confirmed by ^13^C APT NMR, where the piperazine moiety showed one methylene signal in the range of δ 48.66–46.81 ppm. The 1,3,4-thiadiazole carbon atoms appeared as two quaternary carbons in ^13^C APT NMR at a range of δ 184.46–128.27 ppm based on the substituent attached to 1,3,4-thiadiazole moiety.

Additionally, compound **2** showed a broad exchangeable signal corresponding to two hydrazino protons in ^1^H NMR at δ 4.44 ppm. Furthermore, ^1^H NMR of derivative **4** showed a singlet at δ 2.69 ppm for the two methyl groups, which appeared at δ 26.12 ppm in ^13^C APT NMR, whereas compound **5** had a broad exchangeable signal at δ 13.57 ppm, attributed to the thiol proton. ^1^H NMR of bis(thioether) **6a–d** showed either a doublet and two triplets, as in the unsubstituted derivative **6a,** or two doublets, as in *p*-substituted derivatives **6b–d**, corresponding to the phenyl moiety. ^1^H NMR of compound **6d** contained another signal at δ 2.73 for the two equivalent CH_3_ protons, which appeared in ^13^C APT NMR as a primary carbon atom at δ 20.70 ppm. ^1^H NMR spectra of acid products **7a,b** and **8** showed a broad exchangeable signal for hydroxyl groups at δ 11.06, 14.26, and 13.73 ppm, respectively. ^13^C APT NMR showed signals at δ 193, 181, and 189 ppm, corresponding to their carbonyl carbon for **7a,b** and **8,** respectively, whereas, acid **8** had singlet δ 3.15 attributed to two methylene groups of bis(thioglycolic acid) moiety. The proton NMR spectrum of bisaniline **9** showed a broad exchangeable signal at δ 13.57 ppm for two NH protons, while morpholino derivative **10** revealed two signals at δ 3.65 and 2.99 ppm for piperazine and morpholine protons, respectively. It may be deduced that the synthesized compounds are stable in solution, since the spectra that were obtained after 12 and 24 h were quite similar to the original spectra.

### 2.2. Antimicrobial Studies

All the synthesized compounds **2**–**10** were tested for their antimicrobial activity against *S. aureus* and *B. subtilis* as species of gram-positive bacteria and *E. Coli* and *P. vulgaris* as species of gram-negative bacteria. They were also tested against fungi like *C. albicans* and *A. flavus* using the Agar Well Diffusion technique [59,60]. Gentamycin and Ketoconazole were used as antibacterial and antifungal reference drugs, respectively. Dimethyl sulfoxide (DMSO) was used as negative reference. The antimicrobial results in Table 1 indicate that the tested compounds **2**–**10** showed significant antibacterial activity against gram-negative strains, especially *E*. *coli***,** relative to gram-positive bacteria, whereas, they showed weak antifungal activity. Compound **4** showed the highest activity against *S. aureus* with inhibition zone 38 mm, while compound **6d** had the highest activity against *B. subtilis.* In a similar way, with respect to gram-negative bacteria, almost all tested compounds **2**–**10** had higher activity than the gentamycin reference for *E*. *coli*. All compounds had moderate antifungal activity; only compound **6b** showed more activity than the reference used.

The minimum inhibitory concentration (MIC) as an additional test was applied to measure which compounds have good antimicrobial activity. It is well known that MIC is the lowest concentration causing full inhibition of the tested microorganism’s growth. The MIC were determined via the double dilution technique [61]. Six concentrations were prepared (8, 16, 32, 64, 128, and 256 μg/mL) for each tested compound. MIC values were determined by comparison to gentamycin as reference drug, and the MIC data of the tested compounds **2**–**10** are shown in Table 2. The table shows that compounds **4, 6c**, and **6d** exhibited the highest antibacterial activity against *S. aureus,* with MIC 16 μg/mL. On the other hand, compounds **6d** and **7b** had the highest antibacterial activity against *B. subtilis,* with MIC 16 μg/mL. Moreover, compound **6c** exhibited the highest antibacterial activity against *E. coli,* with MIC 8 μg/mL.

### 2.3. Molecular Docking Studies

Docking studies were performed to illuminate the significant antibacterial activities of compounds **2**–**10** via the protein–ligand interaction mechanism. Previously, a variety of ENR inhibitors, such as piperazines, triazoles, and thiadiazol derivatives [62,63,64,65,66,67], were reported. To assess the binding interactions of the structurally similar prepared compounds to the enoyl reductase active site, docking was performed using the MOE (2015) for enoyl reductase from *E*. *coli* (PDB: 1C14 [68]).

As previously mentioned, the ENR reduction process is ruled by two active sites of the enzyme, namely Tyr 158 and Lys 163, in addition to the NAD(H) coenzyme present in the enzyme active site. The piperazine derivatives **2**–**10** interact with enoyl reductase amino acids and give very good scores, varying from −6.0934 to −9.9114 kcal/mol; see Table 3 and Figure 2. The molecular interactions showed that the potential drug binding sites of enoyl reductase are TYR 158, ALA 191, GLY 96, THR 196, VAL 65, GLY 192, LYS 165, PHE 149, MET 103, PHE 41, ILE 16, MET 161, ILE 194, MET 199, GLY 14, THR 39, ALA 191, SER 94 and ASP 64. Triclosan (TCL), is a trichlorinated biphenyl ether; TCL can be considered as ENR inhibitor and can prevent the bacterial fatty acid biosynthesis step. TCL as a reference drug was also docked with enoyl reductase amino acids with a score of −5.8149. It displayed hydrogen bonding interaction with the active site Tyr 158. The hydrogen of the hydroxy group of TCL forms hydrogen bonds with OH of the active site TYR 158 (1.8 Å).

In the case of **7a** and **7b**, which comprised five rings, two were terminal phenyl or pyridyl rings, respectively. Moreover, each terminal ring possessed a carboxy group. Each of these compounds showed hydrogen-bonding interaction with the active site Tyr 158. This is in accordance with the good biological activity results and inhibition zone shown by these compounds. However, all the other ligands showed no interaction with the ENR active site Tyr 58 or Lys 163. This suggests that **7a** and **7b** are superior ligands to the other compounds in binding with ENR active sites. Moreover, these findings propose that other ligands may have a different mechanism to explain their antibacterial activities. Figure 3 shows the 2D (left) and 3D (right) binding modes of selected compounds (green tube) in enoyl reductase active sites.

## 3. Experimental

### 3.1. Instruments and Apparatus

Melting points were determined by MEL-TEMP II melting point apparatus in open glass capillaries. The IR spectra were recorded as potassium bromide (KBr) discs on a Perkin-Elemer FT-IR (Fourier-Transform Infrared Spectroscopy), Faculty of Science, Alexandria University, Alexandria, Egypt. The NMR spectra were carried out at ambient temperature (~25 °C) on a (JEOL) 500 MHz spectrophotometer using tetramethylsilane (TMS) as an internal standard; NMR Unit, Faculty of Science, Mansoura University, Mansoura, Egypt. Elemental analyses were performed at the Regional Center for Mycology and Biotechnology, Al-Azhar University, Cairo, Egypt. 

### 3.2. Agar Disk-Diffusion Method

The synthesized piperazines were tested in vitro using the conventional agar disk-diffusion method against gram-positive (*S. aureus, Staphylococcus aureus* ((RCMB010010); *B. subtilis*, *Bacillus subtilis* (RCMB 015 (1) NRRL B-543)), gram-negative (*E. coli, Escherichia coli* (RCMB 010052) ATCC 25955; *P. vulgaris, Proteus vulgaris* (RCMB 004 (1) ATCC 13315)), and fungi strains (*C. albicans*, *Candida albicans* (RCMB 005003 (1) ATCC 10231); *A. flavus, Aspergillus flavus* (RCMB 002002)).

The piperazines were dissolved in DMSO (which has no inhibition activity) to obtain concentrations of 250 mg/L. The test was performed on nutrient agar (NA) or medium potato dextrose agar (PDA). Uniform size filter paper disks of 5 mm (three disks per compound) were impregnated with equal volumes (10 μL) from the specific concentration of dissolved tested compounds and then carefully placed on the inoculated agar plate surface. After incubation for 36 h at 27 °C in the case of bacteria and for 48 h at 24 °C in the case of fungi, inhibition of the organisms (evidenced by a clear zone surrounding each disk) was measured and used to calculate the mean of inhibition zones [53].

### 3.3. Determination of MIC

All the bacteria were incubated and activated at 37 °C for 24 h inoculation into nutrient broth, and the fungi were incubated in malt extract broth for 48 h. The compounds were dissolved in DMSO and then diluted using cautiously adjusted Mueller–Hinton broth. The two-fold serial concentrations dilution method (8, 16, 32, 64, 128, and 256 µg/mL) of some compounds was employed to determine the MIC values. In each case, triplicate tests were performed, and the average was taken as the final reading. The tubes were then inoculated with the test organisms, grown in their suitable broth at 37 °C for 24 h for tested microorganisms (1 × 10^8^ CFU/mL for bacteria and 1 × 10^6^ CFU/mL of fungi); each 5 mL received 0.1 mL of the above inoculum and was incubated at 37 °C for 24 h. 

### 3.4. Docking Program

Molecular docking simulations were performed to achieve the mode of interaction of prepared piperazines with the binding pocket of enoyl reductase. The newly released crystal structure of enoyl reductase as a receptor was retrieved from the protein data bank (www.rcsb.org, accessed on 12 February 2022) with PDB ID:1C14 [68]. Software version 2015.10 of the Molecular Operating Environment (MOE) was used to prepare the input files and analyze the results. All water molecules, ligands, and ions were removed from the pdb file for the preparation of the protein input file. The active site was selected utilizing the ‘Site Finder’ MOE 2015.10 feature. Prior to docking, the piperazines **2**–**10** and gentamycin were subjected to energy minimization and geometry optimization before docking. Docking simulations were conducted several times with various fitting protocols to observe the best molecular interactions and free binding energies. All docking results were sorted by scoring binding energy.

### 3.5. Synthesis of Compounds ***2**–**10***

#### 3.5.1. 1,4-Bis(5-hydrazinyl-1,3,4-thiadiazol-2-yl)piperazine **2**

A mixture of **1** (3.5 gm, 11 mmol) and hydrazine hydrate (98%, 2 mL) in ethanol (20 mL) was refluxed for 5 h. On cooling the reaction mixture, a solid was separated, which was crystalized from ethanol as white powder (90%), m.p. 290 °C. IR (KBr) υ: 3296 (NH_2_), 3146 (NH), 2941 (sp^3^ -C-H), 1510 (C=N), 1138 (C-N) and 735 (C-S) cm^−1^. ^1^H NMR (500 MHz, DMSO) δ: 4.44 (Br.s, 6H, 2NH and 2NH_2_, D_2_O exchangeable), 3.77 (s, 8H, piperazine-4CH_2_) ppm. ^13^C APT NMR (126 MHz, DMSO) δ: 182.22, 166.30, 48.25 ppm. C_8_H_14_N_10_S_2_ requires: C: 30.55; H: 4.49; N: 44.55%, found: C: 30.12; H: 4.65; N: 44.89%.

#### 3.5.2. 1,4-Bis(5-chloro-1,3,4-thiadiazol-2-yl)piperazine **3**

Thionyl chloride (15mL) was added dropwise with constant stirring to **1** (0.5 gm, 1.5 mmol) for 1 h. The mixture was then refluxed for 5 h, followed by distillation using dry benzene to remove excess of thionyl chloride. The formed product was filtered and crystalized from methanol as pale orange powder (99%); m.p. 240 °C. IR (KBr) υ: 2898 (sp^3^C-H), 1548 (C=N), 1096 (C-N) and 719 (C-S) cm^−1^. ^1^H NMR (400 MHz, DMSO) δ: 3.40 (s, 8H, piperazine-4CH_2_) ppm. ^13^C APT NMR (101 MHz, DMSO) δ: 171.40, 128.27, 48.66 ppm. C_8_H_8_N_6_S_2_Cl_2_ requires: C: 29.72; H: 2.49; N: 26.00%, found: C: 30.05; H: 2.22; N: 26.35%.

#### 3.5.3. 1,4-Bis(3-methyl-1,2,4-triazolo[3,4-b]-1,3,4-thiadiazol-6-yl) piperazine **4**

A mixture of **2** (0.25 g, 0.7 mmol) and acetic anhydride (10 mL) was refluxed for 25 h. On cooling the precipitate was filtered, dried, and recrystallized from ethanol as light yellow powder (86%); m.p. 220 °C. IR υ cm^−1^ (KBr): 2913 (sp^3^ C-H), 1561 (C=H), 1173 (C-N) and 719 (C-S). ^1^H NMR (400 MHz, CDCl_3_) δ: 3.66 (s, 8H, piperazine-4CH_2_), 2.69 (s, 6H, 2CH_3_) ppm. ^13^C APT NMR (101 MHz, DMSO) δ: 181.43, 174.38, 173.15, 46.91, 26.12 ppm. C_12_H_14_N_10_S_2_ requires: C: 39.75; H: 3.90: N: 38.65%, found: C: 39.52; H: 3.77: N: 38.43%.

#### 3.5.4. 6,6′-(Piperazin-1,4-diyl)bis(1,2,4-triazolo[3,4-b]-1,3,4-thiadiazole-3-thiol) **5**

A mixture of **2** (0.25 gm, 0.7 mmol) and carbon disulphide (0.5 mL) in pyridine (10 mL) was refluxed until the evolution of H_2_S ceased. The reaction mixture was then poured onto ice, and the precipitate was collected by filtration and dried as pale yellow crystals, (67%); m.p. 225 °C. IR (KBr) υ: 2910 (sp^3 -^C-H), 2575 (SH), 1554 (C=N), 1172 (C-N) and 717 (C-S) cm^−1^. ^1^H NMR (400 MHz, DMSO) δ: 13.57 (Br.s, 2H, 2SH, D_2_O exchangeable), 3.39 (s, 8H, piperazine-4CH_2_) ppm. ^13^C APT NMR (101 MHz, DMSO) δ: 181.63, 172.47, 163.06, 46.81 ppm. C_10_H_10_N_10_S_4_ requires: C: 30.13; H: 2.53; N: 35.15%, found: C: 29.88; H: 2.26; N: 35.22%.

#### 3.5.5. 1,4-Bis(5-(thioaryl)-1,3,4-thiadiazol-2-yl)piperazine **6a**–**d**

##### General Procedure

An methanolic solution of **3** (0.25 gm, 0.77 mmol) in methanol (10 mL) was mixed with a methanolic solution of substituted thioaryloxide, namely, thiophenoxide, *p*-nitrothiophenoxide, *p*-bromothiophenoxide, and *p*-methylthiophenoxide. The thioaryloxide was prepared by dissolving sodium (0.035 gm, 1.5 mmol) in (10 mL) methanol, followed by addition of substituted thiophenol (1.5 mol). The reaction mixture was stirred for 1–2 h at room temperature, and the separated solid was filtered and crystalized from methanol.

#### 3.5.6. 1,4-Bis(5-(phenylthio)-1,3,4-thiadiazol-2-yl)piperazine **6a**

Pale yellow crystals, (61%) yield; m.p. 120 °C. IR (KBr) υ: 3053 (sp^2^ =C-H), 2913 (sp^3^ C-H), 1636 (C=N), 1113 (C-N) and 738 (C-S) cm^−1^. ^1^H NMR (400 MHz, DMSO) δ: 7.52 (d, *J* = 7.9 Hz, 4H, Ar-H), 7.39 (t, *J* = 6.9 Hz, 4H, Ar-H), 7.29 (t, *J* = 6.8 Hz, 2H, Ar-H), 3.52 (s, 8H, piperazine-4CH_2_) ppm. ^13^C APT NMR (101 MHz, DMSO) δ: 135.89, 130.34, 129.63, 128.96, 127.76, 127.33, 48.14 ppm. C_20_H_18_N_6_S_4_ requires C: 51.03; H: 3.86, N: 17.85%, found: C: 50.87; H: 3.66, N: 17.52%.

#### 3.5.7. 1,4-Bis(5-(4-nitrophenylthio)-1,3,4-thiadiazol-2-yl)piperazine **6b**

Dark brown powder, (74%); m.p. 125 °C. IR (KBr) υ: 3094 (sp^2^ =C-H), 2908 (sp^3 -^C-H), 1594 (C=N), 1107 (C-N), 736 (C-S), and 1510–1339 (NO_2_, asymmetry and symmetry) cm^−1^. ^1^H NMR (400 MHz, DMSO) δ: 8.22 (d, *J* = 8.5 Hz, 4H, Ar-H), 7.81 (d, *J* = 8.1 Hz, 4H, Ar-H), 3.36 (s, 8H, piperazine-4CH_2_) ppm. ^13^C APT NMR (101 MHz, DMSO) δ: 146.75, 143.74, 130.13, 126.52, 124.62, 114.52, 47.43 ppm. C_20_H_16_N_8_O_4_S_4_ requires: C: 42.84; H: 2.88, N: 19.99%, found: C: 43.04; H: 3.09, N: 19.73%.

#### 3.5.8. 1,4-Bis(5-(4-bromophenylthio)-1,3,4-thiadiazol-2-yl)piperazine **6c**

Pale yellow crystals, (60%); m.p. 100 °C. IR (KBr) υ: 3073 (sp^2^ =C-H), 2905 (sp^3^ -C-H), 1634 (C=N), 1109 (C-N) and 721 (C-S) cm^−1^. ^1^H NMR (400 MHz, DMSO) δ: 7.57 (d, *J* = 7.2 Hz, 4H, Ar-H), 7.45 (d, *J* = 8.5 Hz, 4H, Ar-H), 3.47 (s, 8H, piperazine-4CH_2_) ppm. ^13^C APT NMR (101 MHz, DMSO) δ: 135.00, 134.39, 132.32, 129.30, 122.35, 120.91, 47.82 ppm. C_20_H_16_Br_2_N_6_S_4_ requires: C: 38.22; H: 2.57, N: 13.37%, found: C: 38.02; H: 2.47, N: 13.67%.

#### 3.5.9. 1,4-Bis(5-(4-methylphenylthio)-1,3,4-thiadiazol-2-yl)piperazine **6d**

Pale yellow crystals, (65%); m.p. 250 °C. IR (KBr) υ: 3025 (sp^2^ =C-H), 2912 (sp^3^ -C-H), 1637 (C=N), 1113 (C-N) and 740 (C-S) cm^−1^. ^1^H NMR (400 MHz, DMSO) δ: 7.61 (d, *J* = 8.1 Hz, 4H, Ar-H), 7.42 (d, *J* = 7.9 Hz, 4H, Ar-H), 3.48 (s, 8H, piperazine-4CH_2_), 2.73 (s, 6H, 2CH_3_) ppm. ^13^C APT NMR (101 MHz, DMSO) δ: 139.08, 137.68, 132.70, 131.03, 130.20, 128.27, 48.57, 20.70 ppm. C_22_H_22_N_6_S_4_ requires: C: 52.97; H: 4.45, N: 16.85%, found: C: 52.76; H: 4.57, N: 16.65%.

#### 3.5.10. 2,2′-((Piperazin-1,4-diyl)bis(1,3,4-thiadiazol-5,2-diyl))bis acids **7a**,**b** and **8**

A solution of **3** (0.25 gm, 0.77 mmol) in (10 mL) methanol was mixed with methanolic solution (10 mL) of acid thiols, namely, 2-mercaptobenzoic acid, 2-mercaptonicotinic acid, and thioglycolic acid (1.4 mmol) and triethyl amine (2 mL). The mixture was stirred for 5 h at room temperature and then refluxed in a water bath for an additional 5 h. The reaction mixture was cooled, and the separated solid was filtered and crystalized from ethanol.

#### 3.5.11. 2,2′-((Piperazin-1,4-diyl)bis(1,3,4-thiadiazol-5,2-diyl))bis(sulfanediyl)dibenzoic acid **7a**

Pale yellow powder, (74%); m.p. 160 °C. IR (KBr) υ: 3436 (-OH), 3075 (sp^2^ =C-H), 2986 (sp^3^ C-H), 1664 (C=O), 1590(C=N), 1161 (C-N) and 721 (C-S) cm^−1^. ^1^H NMR (400 MHz, CDCl_3_) δ: 11.06 (Brs, 2H, 2OH, D_2_O exchangeable), 7.89 (d, *J* = 7.9 Hz, 2H, Ar-H), 7.57 (m, 4H, Ar-H), 7.34 (t, *J* = 6.6 Hz, 2H, Ar-H) ppm. ^13^C APT NMR (101 MHz, CDCl_3_) δ: 193.67, 185.91, 172.63, 148.25, 133.45, 129.11, 127.29, 125.61, 124.62, 77.00, 46.02 ppm. C_22_H_18_N_6_O_4_S_4_ requires: C: 47.29; H: 3.25, N: 15.04%, found: C: 47.52; H: 3.44, N: 15.22%.

#### 3.5.12. 2,2′-((Piperazin-1,4-diyl)bis(1,3,4-thiadiazol-5,2-diyl))bis(sulfanediyl)bis(1,2-dihydro-pyridine-3-carboxylic acid) **7b**

Dark yellow crystals, (79%); m.p. 220 °C. IR (KBr) υ: 3437 (-OH), 3066 (sp^2^ =C-H), 2823 (sp^3^ C-H), 1681 (C=O), 1557 (C=N), 1141 (C-N) and 718 (C-S) cm^−1^. ^1^H NMR (400 MHz, DMSO) δ: 14.26 (Brs, 2H, 2OH, D_2_O exchangeable), 8.49 (d, *J* = 6.4, 2H, pyridine-H), 8.15 (d, *J* = 5.9 Hz, 2H, pyridine-H), 7.11 (t, 2H, pyridine-H-5) ppm. ^13^C APT NMR (101 MHz, DMSO) δ: 181.64, 166.93, 158.84, 152.37, 143.91, 139.07, 120.60, 115.14, 46.92 ppm. C_20_H_16_N_8_O_4_S_4_ requires: C: 42.83; H: 2.88, N: 19.98%, found: C: 42.52; H: 2.76, N: 19.70%.

#### 3.5.13. 2,2′-((Piperazin-1,4-diyl)bis(1,3,4-thiadiazol-5,2-diyl))bisthioglycolic acid **8**

Pale yellow crystals, (66%); m.p. 255 °C. IR υ cm^−1^ (KBr): 3441(-OH), 2901 (sp^3^ C-H), 1630 (C=O), 1548 (C=N), 1055 (C-N) and 718 (C-S). ^1^H NMR (400 MHz, DMSO) δ: 13.73 (Br.s, 2H, 2OH, D_2_O exchangeable ), 3.38 (s, 8H, piperazine-4CH_2_), 3.15 (s, 4H, 2CH_2_). ^13^C APT NMR (101 MHz, DMSO) δ: 189.73, 181.72, 163.41, 47.02, 45.95 ppm. C_12_H_14_N_6_O_4_S_4_ requires: C: 33.16; H: 3.25, N: 19.34%, found: C: 33.28; H: 3.43, N: 19.01%.

#### 3.5.14. 5,5′-(Piperazine-1,4-diyl)bis(N-phenyl-1,3,4-thiadiazol-2-amine) **9** and 1,4-bis(5-morpho-lino-1,3,4-thiadiazol-2-yl)piperazine **10**

A mixture of **3** (0.25 gm, 0.77 mmol) was treated with amines, namely, morpholine (0.66 mL) and aniline (0.7 mL) in the presence of triethylamine (2 mL) in acetonitrile (5 mL) and refluxed for 24 h. On cooling, a solid was precipitated, and there was no need for crystallization.

#### 3.5.15. 5,5′-(Piperazine-1,4-diyl)bis(N-phenyl-1,3,4-thiadiazol-2-amine) **9**

Pale yellow crystals, (51%); m.p. 260 °C. IR (KBr) υ: 3443 (N-H), 3074 (sp^2^ =C-H), 2908 (sp^3^ -C-H), 1635 (C=N), 1093 (C-N) and 717 (C-S) cm^−1^. ^1^H NMR (400 MHz, DMSO) δ: 13.57 (Br.s, 2H, 2NH, D_2_O exchangeable), 7.27 (m, 10H, Ar-H), 3.40 (s, 8H, piperazine-4CH_2_) ppm. ^13^C APT NMR (101 MHz, DMSO) δ: 181.85, 170.99, 131.24, 129.17, 128.86, 128.07, 47.46 ppm. C_20_H_20_N_8_S_2_ requires: C: 55.01; H: 4.62, N: 25.67%, found: C: 55.37; H: 4.47, N: 25.43%.

#### 3.5.16. 1,4-Bis(5-morpholino-1,3,4-thiadiazol-2-yl)piperazine **10**

Pale white crystals, (58%); m.p. 160 °C. IR (KBr) υ: 2961 (sp^3^ C-H), 1637 (C=N), 1110 (C-N) and 744 (C-S) cm^−1^. ^1^H NMR (400 MHz, DMSO) δ: 3.65 (m, 16H, piperazine-4CH_2_ and morpholine-2CH_2_-O), 2.99 (t, 8H, and morpholine-2CH_2_-N) ppm. ^13^C APT NMR (101 MHz, DMSO) δ: 182.39, 167.02, 66.44, 55.80, 48.65 ppm. C_16_H_24_N_8_O_2_S_2_ requires: C: 45.25; H: 5.70, N: 26.39%, found: C: 45.12; H: 5.52, N: 26.78%.

## 4. Conclusions

A series of *N*,*N*′-disubstituted piperazines **2**–**10** were prepared, and their chemical structures were assigned by IR, NMR, and elemental analysis. The synthesized compounds **2**–**10** were tested for their antibacterial and antifungal activity. The antimicrobial results indicated that the tested compounds **2**–**10** showed significant antibacterial activity against gram-negative strains, especially *E. Coli*, relative to gram-positive bacteria, whereas they showed weak antifungal activity. Compound **4** showed the highest activity against *S. aureus.* Moreover, compound **6d** had the highest activity against *B. subtilis,* while compound **6b** had considerable antifungal activity. The minimum inhibitory concentration (MIC) showed that compounds **4, 6c**, and **6d** exhibited the highest antibacterial activity against *S. aureus.* On the other hand, compounds **6d** and **7b** had the highest antibacterial activity against *Bacillus subtilis*. Moreover, compound **6c** exhibited the highest antibacterial activity against *E. coli*. Docking analysis supported the experimental biological results. The proper interaction with the active sites of ENR was noticed especially for ligands **7b** and **7a**.

## Data Availability

The data presented in this study are available on request from the corresponding author.

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
