# Peer review of "Synthesis and Antimicrobial Activity Screening of Piperazines Bearing N,N′-Bis(1,3,4-thiadiazole) Moiety as Probable Enoyl-ACP Reductase Inhibitors"

_molecules, 2022, doi:10.3390/molecules27123698_

Round 1

Reviewer 1 Report

I have reviewed manuscript number": molecules-1746657, entitled “Synthesis and Antimicrobial Activity Screening of Piperazines Bearing N,N'-Bis(1,3,4-thiadiazole) Moiety as probable EnoylACP Reductase Inhibitors”. The authors prepared new N,N'-disubstituted piperazines conjugated with 1,3,4-thiadiazole and 1,2,4-triazole. They characterize the compounds by IR, NMR, and elemental analysis. They also explore the antimicrobial activity of the compounds. Overall, the manuscript is written and organized well. After check the whole manuscript, some problems can be seen.

  1. Why is the research important to do?
  2. What properties make the studied compounds peculiar?
  3. The authors must be added the following reference in the antimicrobial activity of the introduction section:

https://doi.org/10.17795/zjrms1010

  1. Authors should comment on the solubility nature of the synthetic compounds in the revised manuscript.
  2. Whether the synthetic molecules stable in solution? They should comment on the molecular stability in the solution phase.
  3. The authors must be added the following reference in the “Molecular Docking Studies” section.

https://doi.org/10.1016/j.molliq.2018.01.029

  1. Some “References” should be corrected based on the “Molecules” format.

For example No. 53-59.

  1. The English language of the manuscript should be carefully checked and some typos should be corrected.

After the authors correct all the above issues, this paper can be published after minor revision.

Author Response

Dear Reviewer,

I would like first to thank the Referee for his valuable and accurate comments.

Response to the Reviewer 1 comments

  1. Why is the research important to do?

This research added a number of new piperazines  to the library of organic chemistry. The synthesized compounds also demonstrated significant antibacterial activity.

  1. What properties make the studied compounds peculiar?

The ease of the synthetic approach (catalyst free methods)

  1. The authors must be added the following reference in the antimicrobial activity of the introduction section:

https://doi.org/10.17795/zjrms1010

It has been addressed

  1. Authors should comment on the solubility nature of the synthetic compounds in the revised manuscript.

It has been addressed

  1. Whether the synthetic molecules stable in solution? They should comment on the molecular stability in the solution phase.

It has been addressed

  1. The authors must be added the following reference in the “Molecular Docking Studies” section.

It has been addressed

https://doi.org/10.1016/j.molliq.2018.01.029

  1. Some “References” should be corrected based on the “Molecules” format.

For example No. 53-59.

Done

  1. The English language of the manuscript should be carefully checked, and some typos should be corrected.

Done

Reviewer 2 Report

I have reviewed the manuscript entitled " Synthesis and Antimicrobial Activity Screening of Piperazines Bearing N,N'-Bis(1,3,4-thiadiazole) Moiety as probable Enoyl-ACP Reductase Inhibitors." This work presents relevant information about the synthesis and identification by IR, NMR and elemental analysis of new N,N'-disubstituted piperazines conjugated with 1,3,4-thiadiazole and 1,2,4-triazole and their antimicrobial activity evaluation supported by docking analysis. The manuscript is attractive with promising findings; however, I suggest some changes before being accepted for publication.

Additional comments:

- In the sentence: “Piperazine derivatives have a wide range of biological activities [9-21]” Not many references are necessary to prove the evidence. I suggest removing some references in this sentence and others that are not necessary as there are many references throughout the manuscript.

- Add the reference in this sentence: This compound was found to be more potent than ciprofloxacin against a panel of Gram-positive and gram-negative bacteria,

- Change scheme by figure

- In both assays, indicate how much % of DMSO was added to dissolve the compounds

- In the Agar Disk-Diffusion Method section, indicate which bacteria and fungi were used and where they were obtained. Do not describe in results (S. aureus, Staphylococcus aureus (RCMB010010); B. subtilis, Bacillus subtilis (RCMB 015 (1) NRRL B-543); E. coli, Escherichia coli (RCMB 010052) ATCC 25955; P. vulgaris , Proteus vulgaris RCMB 004 (1) ATCC 13315, C. albicans, Candida albicans RCMB 005003 (1) ATCC 10231, A. flavus, Aspergillus flavus (RCMB 002002)).

- Potato dextrose agar is a medium used to cultivate fungi and yeast. Why was it used for bacteria?

- In the determination of the MIC, please indicate how the MIC was determined.

- Add the unit mm in the sentence: “Compound 4 showed the highest activity against S. aureus with inhibition zone 38”

- It is mentioned that: “other ligands may have different mechanisms to explain their antibacterial activities”. What could those mechanisms be?

- The interaction of the ligands in other sites (not the active site) could have an allosteric effect on the enzyme and present antimicrobial activity?

- Could these compounds interact with the bacterial membrane?

- The conclusion seems more like a summary of results; I suggest making a conclusion considering the impact of the study and highlighting the discovery obtained with the investigation.

- In general, I suggest reviewing and adapting the writing and presentation of the manuscript so that it is in better shape.

Author Response

Dear Reviewer,

I would like first to thank the Referee for his valuable and accurate comments that helped us to revise our manuscript more thoroughly.  All his suggestions have been considered in the revised manuscript in red color.

Response to the Reviewer 2 comments

- In the sentence: “Piperazine derivatives have a wide range of biological activities [9-21]” Not many references are necessary to prove the evidence. I suggest removing some references in this sentence and others that are not necessary as there are many references throughout the manuscript.

Done

- Add the reference in this sentence: This compound was found to be more potent than ciprofloxacin against a panel of Gram-positive and gram-negative bacteria,

Done

- Change scheme by figure

Done

- In both assays, indicate how much % of DMSO was added to dissolve the compounds

Has been addressed

- In the Agar Disk-Diffusion Method section, indicate which bacteria and fungi were used and where they were obtained. Do not describe in results (S. aureus, Staphylococcus aureus (RCMB010010); B. subtilis, Bacillus subtilis (RCMB 015 (1) NRRL B-543); E. coli, Escherichia coli (RCMB 010052) ATCC 25955; P. vulgaris , Proteus vulgaris RCMB 004 (1) ATCC 13315, C. albicans, Candida albicans RCMB 005003 (1) ATCC 10231, A. flavus, Aspergillus flavus (RCMB 002002)).

Has been addressed

- Potato dextrose agar is a medium used to cultivate fungi and yeast. Why was it used for bacteria?

The Agar Disk-Diffusion Method has been updated in the revised manuscript

- In the determination of the MIC, please indicate how the MIC was determined.

Done

- Add the unit mm in the sentence: “Compound 4 showed the highest activity against S. aureus with inhibition zone 38”

Done

- It is mentioned that: “other ligands may have different mechanisms to explain their antibacterial activities”. What could those mechanisms be?

- The interaction of the ligands in other sites (not the active site) could have an allosteric effect on the enzyme and present antimicrobial activity?

- Could these compounds interact with the bacterial membrane?

The alternative mechanism which explain the activity of the other ligands  hasn’t been explored during this study.

- The conclusion seems more like a summary of results; I suggest making a conclusion considering the impact of the study and highlighting the discovery obtained with the investigation.

Done

Reviewer 3 Report

In the present article, the authors describe the synthesis of piperazines containing N,N'-bis(1,3,4-thiadiazole) moiety, screening of their antimicrobial activity, and molecular docking study.

Overall, the manuscript is well done. The synthetic part of the manuscript is sound, with a good presentation of the obtained results and structural characterization of the synthesized compounds. The biological part of the manuscript is also solid, and several derivatives have promising biological activity compared to the corresponding reference compound.

In general, I find only minor issues with regard to the present version of this manuscript.

- The molecular docking study should be slightly more developed, regarding the SAR study for the derivatives analyzed, the possible interactions of their functional groups with the active site and the difference with the interactions found.

- The manuscript should be thoroughly checked for spelling and grammar errors.

I consider this article to be suitable for publication in Molecules, after a minor revision.

Author Response

Dear Reviewer,

I would like first to thank the Referee for his valuable and accurate comments.

Response to the Reviewer 3 comments

The molecular docking study should be slightly more developed, regarding the SAR study for the derivatives analyzed, the possible interactions of their functional groups with the active site and the difference with the interactions found.

Thanks for the suggestion, authors will take this in consideration in future. Probably detailed structure activity relationship will be studied. Which will enhance the prove of the mechanism of action.

- The manuscript should be thoroughly checked for spelling and grammar errors.

Done
